# Aquatic Environments as Hotspots of Transferable Low-Level Quinolone Resistance and Their Potential Contribution to High-Level Quinolone Resistance

**DOI:** 10.3390/antibiotics11111487

**Published:** 2022-10-27

**Authors:** Claudio D. Miranda, Christopher Concha, Félix A. Godoy, Matthew R. Lee

**Affiliations:** 1Laboratorio de Patobiología Acuática, Departamento de Acuicultura, Universidad Católica del Norte, Coquimbo 1780000, Chile; 2Centro i~mar, Universidad de Los Lagos, Puerto Montt 5480000, Chile

**Keywords:** antimicrobial resistance, quinolones, *qnr* genes, PMQR, aquatic environments, low-level resistance

## Abstract

The disposal of antibiotics in the aquatic environment favors the selection of bacteria exhibiting antibiotic resistance mechanisms. Quinolones are bactericidal antimicrobials extensively used in both human and animal medicine. Some of the quinolone-resistance mechanisms are encoded by different bacterial genes, whereas others are the result of mutations in the enzymes on which those antibiotics act. The worldwide occurrence of quinolone resistance genes in aquatic environments has been widely reported, particularly in areas impacted by urban discharges. The most commonly reported quinolone resistance gene, *qnr*, encodes for the Qnr proteins that protect DNA gyrase and topoisomerase IV from quinolone activity. It is important to note that low-level resistance usually constitutes the first step in the development of high-level resistance, because bacteria carrying these genes have an adaptive advantage compared to the highly susceptible bacterial population in environments with low concentrations of this antimicrobial group. In addition, these genes can act additively with chromosomal mutations in the sequences of the target proteins of quinolones leading to high-level quinolone resistance. The occurrence of *qnr* genes in aquatic environments is most probably caused by the release of bacteria carrying these genes through anthropogenic pollution and maintained by the selective activity of antimicrobial residues discharged into these environments. This increase in the levels of quinolone resistance has consequences both in clinical settings and the wider aquatic environment, where there is an increased exposure risk to the general population, representing a significant threat to the efficacy of quinolone-based human and animal therapies. In this review the potential role of aquatic environments as reservoirs of the *qnr* genes, their activity in reducing the susceptibility to various quinolones, and the possible ways these genes contribute to the acquisition and spread of high-level resistance to quinolones will be discussed.

## 1. Introduction

Quinolones belong to a class of synthetic antimicrobial agents with a broad spectrum of activity [1,2,3] and are widely used mostly in the treatment of infections caused by Gram-negative bacteria, and currently remain an important therapeutic option in human and animal medicine [4,5].

During bacterial DNA replication and transcription, double-stranded DNA is uncoiled into a single-stranded structure by enzymes called DNA gyrase or DNA topoisomerase. The antimicrobial activity of quinolones is based on the formation of ternary complexes between DNA and type II topoisomerases (DNA gyrase and topoisomerase IV), which inhibit these enzymes preventing bacterial DNA synthesis, causing cell death [6,7,8]. Gyrase is an excellent target for quinolones because it is not present in eukaryotic cells but is essential for bacterial growth. However, due to the extensive use of these antimicrobials in human and animal therapy, there has been an important increase in the resistance to these drugs [9,10].

The aquatic environment is highly important in the transfer and maintenance of bacterial genes encoding for antibiotic resistance and could serve as a nexus for bacteria inhabiting different environments, such as clinical and aquaculture settings [11,12,13,14]. Therefore, it is reasonable to infer that the aquatic environment and the animals that inhabit it contribute significantly to the persistence and dissemination of quinolone-resistance genes, generating a threat to the efficacy of antimicrobial therapies in human and animal health given the common usage of this class of antimicrobials [15]. Most of the reviews that analyze the presence of genes encoding for antimicrobial resistance in aquatic environments do not commonly include the *qnr* genes [16,17], possibly because they do not confer high-level resistance to quinolones and are not frequently detected among quinolone-resistant bacteria, despite the fact that they play a very important role as precursors for the acquisition of high-level resistance to quinolones, as previously discussed [18].

In this review, the role of aquatic environments as reservoirs of *qnr* genes, their activity in reducing the susceptibility to various quinolones, and the ways these genes contribute to the acquisition and spread of high-level resistance to quinolones will be discussed.

## 2. Quinolone Residues in Aquatic Environments

Antimicrobials are released into aquatic environments via the disposal of urban sewage, hospital wastewater, animal and agricultural waste, effluents from wastewater treatment plants, among other sources [19,20,21]. Consequently, antimicrobial usage and disposal of effluents have led to the aquatic environment in general receiving antimicrobial residues and antimicrobial-resistant bacteria [22,23]. Due to the extensive usage of fluoroquinolones, the presence and accumulation of fluoroquinolones in waters and sediments of both freshwater and marine origin have been widely reported [24]. 

Due to the extensive usage of fluoroquinolones, the presence and accumulation of fluoroquinolones in waters and sediments of both fresh and marine origin have been widely reported as shown in Table 1. Sources of fluoroquinolones mostly include wastewater-treatment plants, agricultural runoff, hospital effluent, and landfill leachate [23,25]. However, residues of several broad-spectrum fluoroquinolone antibiotics have been also extensively documented in the aquatic environment, mostly in surface water, sediment and fish (Table 1).

Fluoroquinolone concentrations in marine waters and sediments are significantly lower compared to freshwater, although marine sediments have a higher affinity towards fluoroquinolone compounds [25,60]. Furthermore, the strong binding of fluoroquinolones to sediment components delays their biodegradation and explains their recalcitrance [67], and their unspecified adsorption to dissolved organic matter causes a decline in the degradation, as was demonstrated for the case of enrofloxacin [68].

Sources of fluoroquinolones mostly include wastewater-treatment plants, agricultural runoff, hospital effluent, and landfill leachate [22]. Fluoroquinolone concentrations in marine waters and sediments are significantly lower than those in freshwater, although marine sediments have a higher affinity towards fluoroquinolone compounds [60]. High levels of several broad-spectrum fluoroquinolone antibiotics were previously documented in effluent, surface water and sediment, ciprofloxacin being the most abundant [26,27,60,69]. Kristiansson et al. [59] showed that river sediments upstream and downstream from an Indian treatment plant contained important levels of fluoroquinolone residues. More recently, Huang et al. [45] found that fluoroquinolone concentrations in water, sediment, and edible fish ranged from 3.49–660.13 ng/L, 1.03–722.18 μg/kg, and 6.73–968.66 μg/kg, respectively. Liu et al. [70] also reported that over 20 different antibiotics were detected in China’s aquaculture sector, with the most common being fluoroquinolones, followed by sulfonamides and macrolides. More specifically, the antibiotics with the highest concentrations found in aquatic products were ciprofloxacin, norfloxacin, and sulfisoxazole, with mean values ranging from 22.5 to 27.9 µg kg^–1^ wet weight.

Despite growing concerns about quinolone resistance, this phenomenon has not been fully explored in environmental settings, possibly because antimicrobial concentrations in non-clinical settings are typically very low [71,72]. On the other hand, antibiotics are naturally produced by environmental microbiota, though in concentrations much lower than those used in antibiotic therapy [73]. Recent studies have revealed that sub-inhibitory concentrations of antibiotics, similar to those found in some aquatic environments, may promote antibiotic resistance and select for resistant bacteria [74,75].

The selective pressure produced by the intensive disposal of pharmaceutical wastes into aquatic environments that can lead to the emergence and maintenance of antimicrobial-resistant bacteria in coastal waters has been documented [76,77,78]. Additionally, antimicrobial residues in these discharges are generally accompanied by high levels of resistant bacteria, most of which carry genes that code for antimicrobial resistance [23]. Previously, anthropogenic factors have been identified as the major drivers of the shift in antimicrobial-resistance gene profiles of aquatic bacteria [79]. Furthermore, a high level of similarity between the antibacterial resistance genes carried by bacteria from hospital environments, and those from the aquatic environments, including those impacted by aquaculture activities, have been observed [72,80].

## 3. Mechanisms of Quinolone Resistance

There are three main mechanisms of quinolone resistance. Firstly, modifications in the enzymes targeted by the drug as a result of chromosomal mutations that result in amino acid substitutions in the quinolone resistance-determining regions (QRDRs) of the GyrA (gyrase) and/or ParC (topoisomerase IV) proteins of Gram-negative bacteria, described as the main target of quinolone action [81,82,83]. Secondly, chromosomal mutations that lead to reduced drug accumulation as a result of either decreased uptake or increased efflux of quinolones [81,82,83]. Thirdly, the presence of quinolone resistance genes that produce either protection of the target enzymes, drug modifying enzymes, or drug efflux pumps [10,84]. These genes are mostly inserted in transferable plasmids, and commonly termed plasmid-mediated quinolone resistance (PMQR) genes [85,86,87,88].

It is well known that efflux-mediated resistance to quinolones and many other antimicrobials is widespread [89,90]. Various multidrug efflux pumps are able to reduce the susceptibility to quinolones and fluoroquinolones in bacteria. There are several families of multidrug-resistant transporters (MDR) in prokaryotes, such as the MFS superfamily (Major Facilitator Superfamily), the ABC superfamily (ATP Binding Compound), the RND family (Resistance Nodulation Division), the MATE family (Multidrug and Toxic Compound Extrusion) and the SMR family (Small Multidrug Resistance) [91]. The effects of efflux pumps belonging to the ABC, MATE, MFS and RND families on reducing the activity of quinolones; mainly ciprofloxacin and norfloxacin have been extensively documented in the clinical environment. Most of these pumps are chromosomally encoded and produced by Gram-positive bacteria [92,93,94,95]. The broad substrate profiles of these pumps link quinolone resistance with multidrug resistance and constitute mechanisms by which the use of non-quinolone antimicrobials may also increase quinolone resistance [90].

### 3.1. High-Level Resistance to Quinolones

The most common mechanism of high-level resistance is due to chromosomal mutations within a short DNA sequence known as the quinolone resistance-determining regions (QRDRs) in one or more of the genes that encode the primary and secondary targets of these drugs, the type II topoisomerases (*gyrA*, *gyrB*, *parC* and *parE*) for both Gram-negative and Gram-positive organisms [3,10,83]. Mutations in the QRDR of these genes, resulting in amino acid substitutions, could alter the target protein structure and subsequently the binding efficiency of the fluoroquinolones, leading to drug resistance [9,10,83]. These resistance mechanisms cannot be transmitted to other bacteria [83]. DNA gyrase has been described as the main target of quinolone action in Gram-negative bacteria [10,96,97]. Thus, the presence of mutations in this enzyme is a critical factor in the emergence of high-level resistance [7,8,10]. Alteration of the primary target site can be followed by secondary mutations in lower-affinity binding sites and highly resistant organisms will typically carry a combination of mutations within *gyrA* and *parC* in Gram-negative organisms [10,83].

Consistent with previous studies, low-level fluoroquinolone resistance in many bacterial species is associated with a single alteration in the GyrA protein, while high-level resistance usually requires at least double mutations [1,2,81]. Single target-site gene mutations typically result in an 8- to 16-fold increase in resistance, with mutation in both DNA gyrase and topoisomerase IV generally associated with higher (up to 100-fold) resistance levels [81,83]. An accumulation of mutations in one or both target enzymes has been shown to cause increasing levels of quinolone resistance, thus it is generally accepted that high levels of quinolone resistance require double *gyrA* mutations [81]. The presence of a single mutation in the above-mentioned positions of the QRDR of *gyrA* usually results in high-level resistance to nalidixic acid, but to obtain high levels of resistance to fluoroquinolones, the presence of additional mutations in *gyrA* and/or in another target such as *parC* is required [98]. Double mutations of *gyrA*–*parC* or *gyrA*–*parE*, and triple mutations of *gyrA*–*parC*–*parE* have been mostly associated with high-level resistance to ciprofloxacin in *P. aeruginosa* isolates [99]. 

Resistance mutations in the GyrB and ParE subunits are considerably less frequent than those in GyrA and ParC [3,10], and appear to be not relevant in the development of quinolone resistance in clinical isolates of Gram-negative microorganisms [81,83,89], only increasing the MIC of quinolones in the presence of a concomitant mutation in *gyrA* [89,90]. Most probably, a high-level of resistance is only observed as a consequence of a synergistic activity with efflux pumps. In general, mutations affecting quinolone uptake and efflux cause only low-level resistance (about four- to eightfold increases in inhibitory concentrations) and do not usually represent a major clinical problem in the absence of additional resistance mechanisms [100]. However, efflux systems have been shown to be of critical importance for the development of high levels of quinolone resistance because it has been reported that quinolone resistance often results from the combination of target alteration and active efflux pumps in enteric bacteria [100,101,102]. Furthermore, the inactivation of major efflux systems prevents the selection of fluoroquinolone-resistant mutants and strains carrying specific target site mutations are no longer clinically resistant if efflux pumps are inactivated [83].

### 3.2. Acquired Low-Level Resistance to Quinolones

The “resistome” concept was introduced by D’Costa et al. [103], and it is defined as the set of resistance determinants in a particular context, such as an aquatic ecosystem [104]. Environmental resistomes have a role as reservoirs of antimicrobial resistance genes, including plasmid-mediated quinolone resistance (PMQR) [11,80,105], which could evolve and spread to human and animal pathogens, jeopardizing the efficacy of antimicrobial therapies, and thus becoming a serious threat for human and animal health [106,107,108].

Plasmid-mediated quinolone resistance (PMQR) genes are mediated by several genes mainly inserted in plasmids and include the *qepA* and *oqxA* genes that encode for active efflux pumps belonging to the superfamilies of transport proteins MFS and RND, respectively, which have been described as conferring a reduction in the susceptibility to nalidixic acid, ciprofloxacin and norfloxacin [109,110]. In addition, efflux pumps encoded by plasmid-located genes *qepA*, *oqxA* and *oqxB*, belonging to the PMQR genes group, have been detected in *E. coli* isolates [110,111,112,113,114,115]. Additional genes include the *aac(6′)-Ib-cr* gene, which encodes for an aminoglycoside acetyltransferase that confers a reduced susceptibility to ciprofloxacin and norfloxacin [109,116], and the plasmid-borne quinolone resistance genes, referred to as *qnr*, which encode for the Qnr proteins that protect DNA gyrase and topoisomerase IV from quinolones, thus preventing their activity [82,109]. Mechanisms of action against quinolones of the PMQR genes are shown in Figure 1.

At present, the *qnr*-determinants are the most frequently detected genes in clinical and environmental bacteria and are distributed across a large number of bacterial genera, principally the Gram-negative bacteria but also to a lesser extent the Gram-positive bacteria [117]. The most effective quinolone resistance mechanisms are the chromosomal mutations that alter the quinolone target proteins and their drug-binding affinity, commonly conferring high levels of resistance to quinolones [118,119], whereas PMQR genes only provide a reduced susceptibility to quinolones [81,82]. Thus, PMQR genes are not able to confer a clinically significant quinolone resistance, but they help reduce the susceptibility to quinolones in bacteria and thus facilitate the selection of mutants with a higher level of resistance [89]. PMQR determinants generally confer only low-level quinolone resistance that alone does not exceed the clinical breakpoint for resistance, thus *qnr*-positive strains remain susceptible to fluoroquinolones. In contrast, a high level of resistance to quinolones implies an increase of MIC values above the clinical breakpoint, resulting in a quinolone-based therapy failure.

When the antimicrobial resistance of environmental bacteria is studied, low-level resistance is not commonly detected using standard susceptibility testing procedures, considering that no clinical breakpoints are stated for most of the environmental non-human pathogenic species. Thus, most of the studies in which *qnr* genes are detected in bacterial isolates typically report quinolone resistance at a clinical level, which most likely indicates simultaneous topoisomerase mutations or an overexpression of efflux pumps [120,121,122]. More recently, studies that investigate the occurrence of *qnr* genes have increasingly been carried out using non-culturable approaches, such as metagenomic analyses. Thus, detection of *qnr* genes in bacterial populations exhibiting only low susceptibility but without quinolone resistance is unusual considering that these strains do not represent an important human health risk.

Currently, various studies have demonstrated that the prevalence of plasmid-mediated quinolone resistance, mainly encoded by the *qnr* genes, such as the *qnrA*, *qnrB* and *qnrS* genes, is widespread in clinical settings and mainly described for the enteric bacteria *Citrobacter*, *Enterobacter*, *Klebsiella* and *Salmonella* [109,123,124], but studies of the prevalence of these genes in aquatic environments are still limited. However, a critical issue still to be solved is that most of the studies only investigate antimicrobial-resistant isolates and discard those exhibiting a reduced susceptibility, which are only considered when they are exhibiting other resistance mechanisms. In addition, most of these studies, which usually aim to detect these genes but not efflux pumps or mutations, typically characterize quinolone resistance in bacteria from clinical and environmental settings only.

It can be concluded that *qnr* occurrence among environmental bacteria should not be unexpected, considering that a continuous bacterial load and antimicrobial residues at sub-inhibitory concentrations are commonly discharged into aquatic environments [19,20]. Furthermore, the intensive worldwide use of antimicrobials in aquaculture, including quinolones and fluoroquinolones such as oxolinic acid, flumequine, norfloxacin and enrofloxacin, which are currently approved and used for this purpose in many countries, such as China, Thailand, Philippines, Vietnam, Chile, Italy and Norway, amongst others [70,125,126,127], has also contributed to the aquatic environment being considered as a reservoir of the quinolone-encoding resistance genes [11,60,80,105].

It should be noted that sub-inhibitory concentrations of quinolones have been reported in sediments near sites impacted by the aquaculture, indicating a clear effect of the currents in the dilution of antimicrobials in the aquatic environment [128,129]. This fact favors the occurrence of *qnr* genes among bacteria associated with aquaculture settings or near fish farms. The lack of knowledge on the persistence of antibiotics in sediments impacted by aquaculture activities as well as the prevalence of antibiotics in trace concentrations in these environments is a crucial but still an unsolved problem, indicating the urgent necessity of understanding the role of aquaculture in the selection of low-level resistance.

The occurrence of quinolones at sub-inhibitory concentrations promotes the selection of *qnr*-carrying strains exhibiting resistance to low concentrations of quinolones, thus allowing them, after successive generations, to develop mutations in the quinolone targets, DNA gyrase and topoisomerase IV and thereby gain a high level of quinolone resistance [18,75,87,88,130]. In addition, the increase in the appearance of antimicrobial-resistant mutants as a consequence of bacterial stress has been observed [131]. Thus, it could be concluded that aquatic environments under starvation conditions could trigger the production of stress signals, previously described as promoting increases in mutation rates [132]. Moreover, low antibiotic concentrations are not only able to select low-level antibiotic-resistant variants but may produce a substantial stress in bacterial populations that eventually influences the rate of genetic variation and the diversity of adaptive responses [18,133].

## 4. Quinolone Resistance Genes (*qnr*)

### 4.1. Origin and Structure of qnr Genes

As has been noted previously, *qnr* genes are the most frequently found PMQR genes, which are increasingly being reported worldwide from pathogenic and non-pathogenic bacteria isolated from both clinical and aquatic environments [81]. Transferable quinolone resistance was reported for the first time in 1998, where a multi-resistant strain of *Klebsiella pneumoniae* that contained a plasmid conferring reduced susceptibility to ciprofloxacin [87,90,109,110], but this transferable resistance, compared to mutations in the quinolone target enzyme, only conferred low-level resistance [89]. The sequencing of the *qnr* gene revealed that this sequence encoded a 218-residue protein in which there were tandem repeats of five amino acids, placing them in the pentapeptide repeat family proteins [89,134]. 

Although the origin of the *qnr* genes has not yet been fully determined, it has been suggested that these proteins come from chromosomal proteins. To date, it is well known that two proteins of the family of the repeating pentapeptides are of relevance in the resistance to quinolones and these maintain 20% homology with QnrA. The first is McbG [134,135], which protects from bacterial self-inhibition of the activity of microcin B17 (MccB17) [86]. MccB17 is a 3.1 kDa post-transcriptionally modified peptide that blocks DNA replication, and like ciprofloxacin, inhibits the action of DNA gyrase and thus the stabilization of the DNA–DNA gyrase complex in the presence of ATP and DNA free ends [86]. The second protein is MfpA, a protein that was cloned from the *Mycobacterium smegmatis* genome in studies related to active ejection pumps that contribute to resistance to quinolones [86,136,137].

The expression of the *mfpA* gene results in as much as a fourfold increase in the MIC to ciprofloxacin [86]. The relationship between the members of this protein family and Qnr is difficult to establish, since the homology between Qnr and McbG or MfpA is 19.6% and 18.9%, respectively [86]. Additionally, it has been speculated that Qnr results from some proteins designed to protect DNA gyrase from natural inhibitors, or from some chromosomal gene of unknown function that encodes a protein of the pentapeptide family from mycobacteria, cyanobacteria or other older bacterial groups [89,110].

Qnr proteins belong to a family of repeated pentapeptides, which bind directly to their targets, the DNA gyrase and Topoisomerase IV enzymes [109]. The protective activity of Qnr has been attributed to its external loops where the elimination of one of them reduces its protective activity while the elimination of both completely eliminates its activity [90,138,139], tandem amino acid mutations also remove the protein’s protection against quinolones [90,140]. Qnr binds to both the gyrase holoenzyme and its A and B subunits, suggesting that Qnr protects gyrase by blocking the access of quinolone to the GyrA sites essential for its action [90].

The high inter- and intra-allelic variability of the *qnr* gene suggests that this gene has undergone successive mutations over time. To date, the alleles *qnrA, qnrB, qnrC, qnrD, qnrE, qnrS* and *qnrVC* [141,142,143,144,145,146,147,148], which are mostly associated with plasmids, have been described, [87,90,121,145,148,149]. Many variants have been detected for the *qnr* allele genes; currently, 14 for QnrA, 96 for QnrB, 1 for QnrC, 3 for QnrD, 4 for QnrE, 15 for QnrS and 10 for QnrVC have been reported, indicating that the *qnrB* allele is the most variable, exhibiting plasmid and chromosomal variants [150,151].

The nucleotide and amino acid sequence identity of the Qnr alleles are shown in Table 2, indicating the highest similarity between QnrB and QnrE alleles (75.81 and 85.98% identity at nucleotide and amino acid level, respectively), and QnrC and QnrVC alleles (68.65 and 73.85% identity at nucleotide and amino acid level, respectively), whereas the lowest amino acid similarities were observed between QnrS allele and the QnrE, QnrD and QnrB alleles with approximately 36, 39 and 40% identity, respectively. According to Hooper and Jacoby [90], the different alleles of *qnr* genes differ by 35% or more between their sequences, whereas qnr allelic variants differ by 10% or less as described in almost all alleles. However, Table 2, which includes representatives of each allele of Qnr, shows that several alleles have nucleotide differences from 24.19% (*qnrB* and *qnrE*) to 53.67% (*qnrA* and *qnrB*).

The *qnr* genes are distributed in a large number of bacterial genera, mainly the Gram-negative bacteria but also to a lesser extent the Gram-positive bacteria [152]. These genes have been described as part of the transferable incompatibility plasmids with an approximate size ranging from 2.7 to 320 kb and with a wide host range, which favors their dissemination [87,88,110]. Interestingly, *qnr* genes have also been described in the Vibrionaceae family and it is suggested that these bacteria could be a natural reservoir for these kinds of resistance determinants [153,154,155,156]. Furthermore, it has been reported that *qnr* genes are located inside unusual integrons associated with *sul1* genes [134,157,158] and it has also been suggested that there is a relationship between the resistance to quinolones and the production of extended-spectrum β-lactamases [159,160,161].

### 4.2. Antimicrobial Activity of qnr Genes

The *qnr* genes act by preventing the antibiotics from binding to the DNA gyrase and Topoisomerase IV enzymes and exerting their bactericidal activity. Thus, they reduce the susceptibility to the quinolones ciprofloxacin, levofloxacin, norfloxacin and nalidixic acid, amongst others in the carrying bacteria [82,109]. Bacterial strains exhibiting a slightly higher minimum inhibitory concentration (MIC) than is common for the susceptible population, are considered to have low-level resistance.

Minimum Inhibitory Concentrations (MIC) assays are commonly used to investigate the effect of *qnr* genes on lowering quinolone susceptibility, mostly by using cloning and the transfer of *qnr* genes to *Escherichia coli* recipients. Inhibitory activity of these genes is usually examined by measuring the difference in quinolone MICs for an *E. coli* strain with and without a *qnr*-bearing plasmid. There are several studies examining the impact of *qnr* allele variants on quinolone MIC values, as described in Table 3.

The increased ability of *qnr* gene alleles to reduce the susceptibility to ciprofloxacin has been reported, with 30- to 66-fold increase in the MIC values observed. A lower reduction in the susceptibility to levofloxacin has also been observed, with 15- to 62-fold increase in the MIC values, with *qnrA* and *qnrS* alleles exhibiting the highest effect, whereas the studied *qnr* alleles have a lower effect on the MICs for norfloxacin (4- to 33-fold increases). Finally, as can be seen in Table 3, the quinolone with the least pronounced effect was nalidixic acid, with zero to twofold (*qnrD*), fourfold (*qnrA*, *qnrB* and *qnrC*), two to fourfold (*qnrS*) and eightfold (*qnrE*) increases in the MIC values), which are in agreement with a previous report [162]. The only exception was the *qnrVC* gene, with a 133-fold increase in the MIC of nalidixic acid (Table 3). In another study, *Escherichia coli* transformants bearing the *qnrVC5* gene alone showed a twofold to a fourfold elevation in the MIC of norfloxacin and ciprofloxacin, and a fourfold to eightfold increase in the MIC of nalidixic acid when compared to untransformed *E. coli* (Table 3). However, when the effect of the *qnrVC5* gene in combination with *aac(6′)Ib-cr* genes was studied, the same increases in the MIC values of nalidixic acid, levofloxacin and ciprofloxacin were observed as those produced by one of the transformants with *qnrVC* alone [156]. Table 3 shows that all *qnr* genes produce a decrease in susceptibility that does not reach the CLSI (Clinical and Laboratory Standards Institute) clinical breakpoint for resistance, thus addressing the question about the clinical importance of *qnr* genes. The answer is that PMQR genes facilitate the selection of higher levels of quinolone resistance [87].

**Table 3 antibiotics-11-01487-t003:** Minimum Inhibitory Concentration (MIC) of various quinolones by the different alleles of the *qnr* gene, using *Escherichia coli* as a model species.

Gene	MIC (µg/mL)				Reference
	CIP	LVX	NFX	NAL	
*qnrA*	0.125 (0.002)	0.5 (0.008)	0.5 (0.015)	NT	[149]
	0.125 (0.002)	NT	0.25 (0.015)	8 (2)	[144]
	0.125 (0.002)	0.5 (0.008)	NT	NT	[121]
	0.25 (0.008)	0.5 (0.015)	NT	16 (4)	[87]
	0.125 (0.002)	0.125 (0.004)	NT	NT	[163]
	0.25 (0.008)	0.5 (0.015)	NT	16 (4)	[109]
	0.25 (0.008)	NT	NT	NT	[164]
*qnrB*	0.125 (0.002)	0.125 (0.008)	0.25 (0.015)	NT	[149]
	0.125 (0.002)	0.125 (0.008)	NT	NT	[121]
	0.25 (0.008)	0.5 (0.015)	NT	16 (4)	[87]
	0.06 (0.002)	NT	NT	8 (2)	[150]
	0.25 (0.008)	0.5 (0.015)	NT	16 (4)	[109]
*qnrC*	0.25 (0.008)	0.25 (0.015)	NT	16 (4)	[109]
*qnrD*	0.06 (0.002)	NT	0.06 (0.015)	4 (2)	[144]
	0.06 (0.008)	NT	NT	4 (4)	[109]
*qnrE*	0.125 (0.002)	0.125 (0.004)	NT	8 (1)	[148]
*qnrS*	0.125 (0.002)	0.5 (0.008)	0.5 (0.015)	NT	[149]
	0.06 (0.002)	NT	0.06 (0.015)	4 (2)	[144]
	0.125 (0.002)	0.5 (0.008)	NT	NT	[121]
	0.25 (0.008)	0.38 (0.015)	NT	16 (4)	[87]
	0.25 (0.008)	0.38 (0.015)	NT	16 (4)	[109]
	0.25 (<0.01)	NT	1 (0.03)	4 (1)	[165]
*qnrVC*	0.5 (0.0075)	NT	NT	16 (0.12)	[166]
	0.5 (0.125)	NT	4 (1)	400 (50)	[156]
	0.25 (0.125)	NT	2 (1)	200 (50)	[156]
*E. coli* SBV	≤1	≤2	NT	≤16	[167]

CIP: Ciprofloxacin; LVX: Levofloxacin; NFX: Norfloxacin; NAL: Nalidixic Acid; NT: Not Tested. In parenthesis MIC values of *E. coli* not-carrying any *qnr* gene. SBV: CLSI Susceptibility Breakpoint Value [167].

It has been reported that in the presence of *qnr* genes, mutations in the *gyrA* and *parC* genes were easily selected producing high levels of quinolone resistance [10,163], which is clinically very relevant considering that the coexistence of both resistance mechanisms increases the level of resistance to quinolones [87,168,169]. Furthermore, Li [170] concluded that *qnr-*bearing strains generate quinolone-resistant mutants at a much higher frequency than *qnr*-free strains. By contrast, it has also been reported that mutations in the *gyrA* gene that confer resistance to quinolones are infrequent in *E. coli* carrying the *qnr* gene [171], suggesting that the quinolone resistance-determining region (QRDR) could be protected from quinolones by the Qnr protein and consequently other mechanisms are required to acquire higher levels of fluoroquinolone resistance, such as the activity of efflux pumps.

### 4.3. Occurrence of qnr Genes in Aquatic Environments

Even though the worldwide information on the occurrence of *qnr* genes in aquatic environments is very scarce, there are an important number of studies reporting their detection, mainly in freshwater environments, indicating their presence in more than 30 countries (Table 4). As observed in Table 4, most of the *qnr* genes are carried by Gram-negative bacteria, with a significant predominance of studies that demonstrate the presence of *qnr* genes in freshwater (78 studies), compared to reports of their detection in the marine environment (22 studies). Furthermore, the predominance of *qnrB* and *qnrS* genes in these environments was observed, along with a very low incidence of the *qnrC* and *qnrVC* genes (Table 4). 

Over the last decade, a significant number of studies have reported the occurrence of *qnr* genes in aquatic environments, as described in Table 4. This is mainly due to the increasing use of molecular tools, such as PCR, quantitative real-time PCR, and metagenomics [222,230,264,265]. Sixty-four out of 100 studies reporting the occurrence of *qnr* genes in aquatic environments were based on the use of molecular methods, with a high number of studies undertaken in Chinese aquatic environments (36 studies).

Despite the fact that there has been a growing increase in *qnr* gene detection studies in aquatic environments, these are mainly limited to their molecular detection, which prevents an understanding of their functionality and potential location in mobile elements. This is necessary for understanding the role of aquatic environments in the dissemination of these genes. Therefore, only the isolation and analysis of bacteria carrying these genes in aquatic environments will allow for the determination of whether they can be horizontally transferred. If this is so, *qnr* genes could constitute an important threat to human and animal health.

As has been previously noted, *qnr* genes are usually located in mobile genetic elements such as transferable plasmids [87,88,110], which could explain why these genes are widely distributed in Gram-negative bacteria, mostly carried by fermenting bacilli belonging to the *Enterobacteriaceae* family [159,163,266,267], as well as by non-fermenting bacilli belonging to the *Pseudomonas* genus, whereas they have only been detected in a small proportion of Gram-positive bacteria, mainly chromosomally located [88,186,222]. However, in aquatic environments, a high taxonomic diversity of bacteria carrying a *qnr* gene has been reported (Table 4).

In Thailand and Vietnam, plasmids that provide fluoroquinolone resistance have been detected in contaminated waters, specifically the *qnrB* gene associated with enteric bacteria [239]. In Chile the presence of *qnrA*, *qnrB* and *qnrS* genes, and the *oqxA* gene encoding for an efflux pump were described in isolates from uncontaminated sediments and sectors near a salmon farm [246,247]. These observations are a clear example of the worldwide distribution in aquatic environments of these genes and a reflection of their impact on animal and human health.

On the other hand, as shown in Table 4, the occurrence of genes associated with low-level quinolone resistance has also been observed in bacteria isolated from wild and farmed fishes in various countries. The presence of the *qnrB, qnrS* and *qnrD* genes have been detected in *Escherichia coli* isolates recovered from fish cultures in China [178]. Plasmid-borne *qnrA, qnrB* and *qnrS* genes were isolated from bacteria associated with fish cultures in Egypt [255]. In addition, the presence of *qnrS* and *qnrB* was detected in strains of *Aeromonas* spp. isolated from fish in South Africa [236], *qnrS* was found in *Aeromonas* spp. isolated from diseased fish from fish farms and aquariums in Korea [219], and the *qnrA* and *qnrD* genes were detected in marine and freshwater animals captured along the Chinese coast, whereas *qnrB*, *qnrC* and *qnrS* were not found [13].

As was stated by Jacoby et al. [268], *qnrB* is the most common of the five *qnr* families and has the greatest number of allelic variants, primarily detected in the *Citrobacter* genus, with several of them located on the bacterial chromosome [62]. This is further supported by the only study of a *qnr* gene detected in farmed fish in Chile, which reported a non-transferable *qnrB* gene carried by a *Citrobacter gillenii* strain recovered from the mucus of farmed salmon [151]. The study by Concha et al. [151] of the genetic environment of the *qnrB* gene carried by the *C. gillenii* strain, as compared to other *qnrB* genes, suggested an environmental origin as the most likely source, rather than a clinical source.

### 4.4. Prevalence and Spread of qnr Genes in Aquatic Environments

The prevalence and spread of *qnr* genes, as well as most of the genes conferring low-level quinolone resistance, are dependent on various biological characters of the bacterium such as cell duplication rate and transfer ability in the aquatic environment. Thus, the carrying of *qnr* genes by bacterial species inhabiting these environments promotes the spread of these genes, favoring gene flow between distant aquatic ecosystems. Furthermore, external factors associated with the concentration of quinolone residues in the environment, mainly sub-inhibitory concentrations, favor the prevalence of *qnr* genes, whereas high concentrations of the antimicrobial could enhance the prevalence of other mechanisms or the co-existence of two or more antimicrobial resistance mechanisms that confer high levels of resistance to quinolones through additive action (Figure 2).

As has been noted, many aquatic environments commonly contain low amounts of antimicrobials, including fluoroquinolones, as a consequence of human activities [23,60,67], which could have a selective effect that leads to the predominance of low-level resistant populations. Moreover, the extensive use of antibiotics in human and veterinary medicine and their subsequent release into the aquatic environment via treated or untreated wastewater discharges are frequent [165]. Recent studies have revealed that sub-inhibitory concentrations of antibiotics, similar to those found in some aquatic environments [19,20], may promote antibiotic resistance and select for resistant bacteria [51,59]. Furthermore, it has been reported that exposure to sub-MIC of quinolones may increase resistance to non-quinolones [51]. As shown in Figure 2, sub-inhibitory concentrations of quinolones select for those bacterial populations carrying a *qnr* gene, favoring their horizontal transfer when they are associated with mobile elements, typically plasmids and integrons, as has been commonly reported [59]. There exists a high possibility that in plasmids carrying a *qnr* gene, the gene could be inserted as an integron, a mobile genetic element capable of harboring a diverse number of genetic cassettes encoding for resistance to various antimicrobials [269] (Figure 2).

**Figure 2 antibiotics-11-01487-f002:**
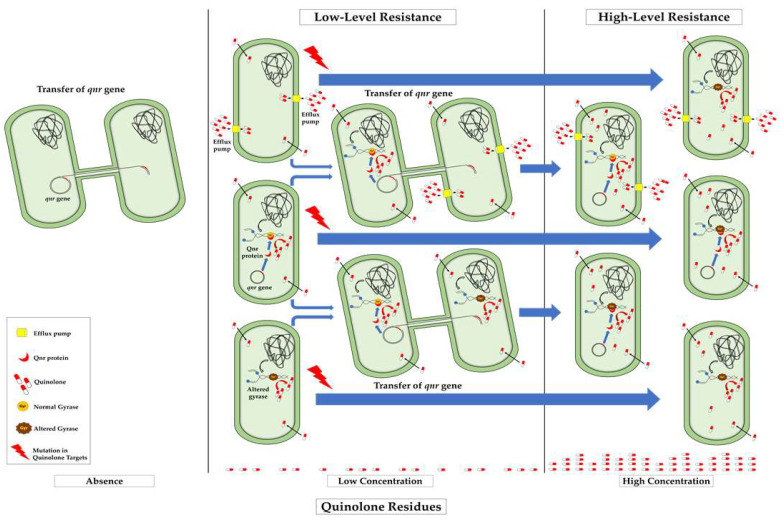
Main events that influence the prevalence and dissemination of *qnr* genes in the aquatic environment and their potential relationships with high-level quinolone resistance. Furthermore, *qnr* genes associated with integrons have frequently been reported [87,157,158,270,271], thus favoring their spread by horizontal transfer. In addition, given the ability of integrons to capture more antimicrobial resistance encoding genes, it is common to find *qnr* genes accompanied by other genes in the same integron, such as *aadA* and *bla* genes, which encode for resistance to aminoglycosides and β-lactams, respectively [23,60]. This situation would make the scenario of antibiotic resistance even more complex.

In another study, several ciprofloxacin-resistant *Citrobacter freundii* recovered from wastewater treatment plants carried a *qnrB* gene as part of a complex integron [158]. These isolates were found to carry mutations in *gyrA* and *parC*, as well as multiple plasmid-borne resistance genes, and the authors suggested the transfer of plasmid-borne fluoroquinolone resistance genes from pathogens to indigenous inhabitants of these environments. This suggests that aquatic environments play a role as a major route for the dissemination of these quinolone resistance determinants [170,272], most probably posing a serious threat to human and animal health.

The horizontal transfer of *qnr* genes inserted in plasmids or integrons enables not only the transfer of these genes but also that other genes encoding for resistance to other antimicrobials, which could be transferred in parallel, depending on the genetic structure of these genetic elements [273,274]. Thus, the dissemination and maintenance of *qnr* genes in bacteria is mainly dependent on the plasmid transfer and the constant selective pressure caused by the sub-inhibitory concentrations of quinolone residues in the surrounding environment. Therefore, the presence of an initial quinolone resistance mechanism can facilitate the acquisition of a second mechanism [18,130,275]. 

The high variability of the *qnr* genes [87,88,148], and the high taxonomic diversity of species carrying those genes in the aquatic environment, suggests that the persistence and dissemination of these genes in these systems are extensive [135]. This increases the possibility that integrons harboring *qnr* genes, and genes encoding efflux pumps, will also have other forms of resistance genes in their structure. Given the transfer and recombination processes undergone over time, this suggests a complex scenario not only for the quinolones but for the overall antibiotic resistance phenomenon.

### 4.5. Role of qnr Genes in the Acquisition of High-Level Resistance

When *qnr* genes are transferred to bacteria containing efflux pumps or topoisomerase mutations, the low quinolone susceptibility of recipients could evolve to a high-level resistance, as a consequence of an additive activity (Figure 2). The presence of a first resistance mechanism may facilitate the acquisition of a second. Furthermore, in previous studies it has been observed that porins and efflux pumps are environmentally regulated and can confer low-level resistance to quinolones [276]. The overexpression of various efflux pumps can lead to low-level resistance [277,278,279], lowering the cytoplasmic concentration of antimicrobials inside the cell and generating an advantage for the evolutionary selection of high-resistance strains [279,280,281]. In support of this trend, the combined activity of *qnr* genes with multidrug efflux systems drastically reducing the susceptibility to quinolones and providing a high-level of quinolone resistance has been reported [122,282,283], confirming the additive inhibitory activity of both antimicrobial mechanisms [18], as postulated in Figure 2.

Bacteria carrying *qnr* genes inhabiting aquatic environments containing low concentrations of quinolones, will most probably be selected with low biological cost, increasing their competitive capability compared to the susceptible population. The exposure to low-level antibiotic concentrations will enable the population to persist for long periods and induce genome instability [284,285,286], allowing for the development or overexpression of other mechanisms, which combined with *qnr* gene expression will provide higher levels of resistance [26,279,287]. Furthermore, a continuous exposure to low amounts of quinolone residues will contribute to the selection of primary mutants exhibiting low-level resistance to quinolones, which could eventually be recipients of *qnr* genes, resulting in high-level resistant bacteria (Figure 2).

Thus, as shown in Figure 2, the synergistic combination of different quinolone low-level resistance mechanisms commonly results in a high-level resistance. It is thus of great concern that the simultaneous occurrence of *qnr* genes and main components of intrinsic resistance, such as the efflux pumps, are exhibited by various bacterial groups [288,289]. This issue is highly important because these bacteria will be able to survive in impacted aquatic environments receiving high levels of quinolone residues.

It has been observed that bacteria carrying mutations in the quinolone action target enzymes can exhibit higher MIC values when efflux systems are expressed [18,282,290]. The same result was observed in *Pseudomonas* isolated from freshwater environments associated with fish farming [283]. Furthermore, it has been reported that enteric bacteria with high and intermediate resistance to quinolones exhibited the co-occurrence of *qnr* genes and mutations in the GyrA and ParC subunits [122]. Additionally, in a study using quinolone-resistant *V. fluvialis*, a twofold to fourfold increase in MIC of ciprofloxacin ofloxacin due to additional presence of the PMQR determinants *qnrVC5* and *aac (6′)-Ib-cr*, apart from GyrA and ParC mutations was reported. These results suggested that *qnrVC5* and *aac (6′)-Ib-cr* determinants may contribute toward resistance to ciprofloxacin [156].

## 5. Conclusions

The increasing number of studies reporting the occurrence of *qnr* genes carried by aquatic bacteria strongly suggests that aquatic environments are important reservoirs of these genes, but also that they are frequently underestimated considering that *qnr* genes are almost exclusively investigated among quinolone-resistant bacteria. Therefore, most of the studies dealing with antibiotic resistance in aquatic environments only consider bacterial isolates exhibiting levels of antimicrobial resistance, but commonly do not include bacteria exhibiting low-level resistance, which have minimum inhibitory concentration (MIC) values slightly higher than those of the susceptible population, but lower than those considered as resistant by CLSI breakpoints. Thus, it could be concluded that the occurrence of quinolone resistance genes is seriously underestimated if only fluoroquinolone-resistant bacteria are considered, given the high incidence *qnr* genes that have been detected worldwide in aquatic environments.

The significant number of studies reporting the occurrence of qnr genes in aquatic environments during the last decade is a consequence of the growing use of non-culturable molecular methodologies. This has made it possible to significantly improve the detection of these genes in these environments. Despite the above, it is evident that the occurrence of *qnr* genes in aquatic environments, especially marine, is unknown or mostly underestimated, so more studies are required to demonstrate their presence in unpolluted and anthropogenically impacted aquatic environments.

The occurrence of *qnr* genes in aquatic environments is most probably caused by the release of bacteria carrying these genes through anthropogenic pollution and maintained by the selective activity of antimicrobial residues discharged into these environments. Considering the high probability that PMQR genes can act in combination with other mechanisms of resistance to quinolones, there is no doubt that their increasingly frequent detection in aquatic environments presents a significant threat to the efficacy of quinolone-based human therapies.

It has been well established that the frequency of quinolone resistance in bacteria from clinical settings is a consequence of quinolone usage, whereas the appearance and significance of low-level resistance mediated by *qnr* genes in environmental bacteria are still uncertain and need to be determined. It can be concluded that aquatic environments are important reservoirs of *qnr* genes, with an important role in the acquisition and spread of *qnr* genes. The presence of *qnr* genes in the aquatic environment could also be used for assessing anthropogenic impact.

## 6. Perspectives

The role of aquatic environments, particularly in areas used for recreation and aquaculture, as potential reservoirs of *qnr* genes must be clarified. This raises the urgent need to develop exhaustive studies that investigate the occurrence of *qnr* genes in coastal waters impacted by effluents, in order to implement adequate measures to prevent their dissemination. The monitoring of aquatic environments for the prevalence of transferable *qnr* genes is of immediate importance in order to evaluate the emergence and spread of these genes in the aquatic environment, as well as to protect human health during recreational activities in anthropogenically impacted surface waters. Additionally, studies are required to compare *qnr* genes and their genetic surroundings, particularly if they are associated with mobile elements from environmental and clinical origins. This would advance our understanding of the flux of these genes and their contribution to the acquisition of high-level resistance to fluoroquinolones in clinical and environmental settings.

To accomplish this, the presence of quinolone residues in trace concentrations, together with the detection of low-level quinolone resistance genes associated with mobile elements, must be comprehensively investigated. A thorough understanding of the epidemiological status of the transferable determinants encoding for quinolone resistance in aquatic environments is urgently required.

## Figures and Tables

**Figure 1 antibiotics-11-01487-f001:**
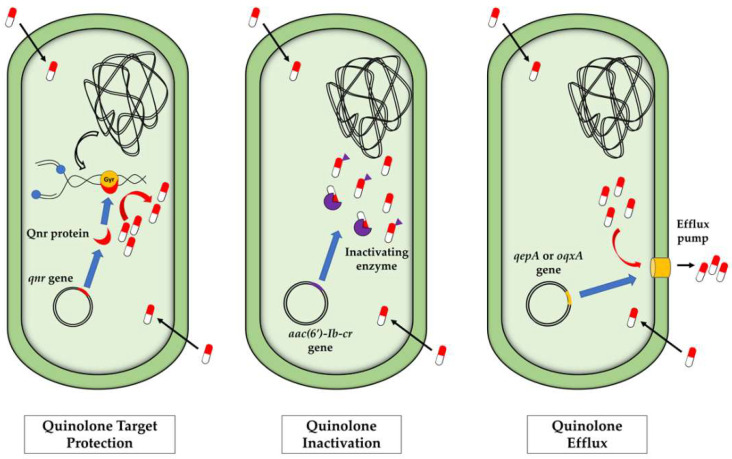
Mechanisms of action of plasmid-mediated quinolone resistance (PMQR) genes.

**Table 1 antibiotics-11-01487-t001:** Detection of residues of various quinolones in aquatic environments.

Sample	Source	Mean or Range of Quinolones (ng/L or µg/kg)				Reference
		OFX	NFX	CIP	ENR	
Water	Suquía River, Argentina	nd–69	nd–80	nd–78	NT	[26]
	Southeast Queensland, Australia	NT	1,150	300	1300	[27]
	Belém River, Brazil	NT	110	<20	NT	[28]
	Barigui River, Brazil	NT	<20–130	70	NT	[28]
	Hunhe River, China	nd–280	nd–1380	nd–65	nd–17	[29]
	Bohai Sea, China	3–5100	32–6800	4.9–390	NT	[30]
	Haihe River, China	180	NT	130	NT	[31]
	Baiyangdian Lake, China	0.38–32.6	nd–156	nd–60.3	nd–4.42	[32]
	Laizhou Bay, China	nd–45.5	nd–572	nd–346	nd–24.6	[33]
	Xiaoqing River, China	9.5–1605	nd	nd–56.6	nd	[34]
	Huangpu River, China	nd–6.5	nd–2.6	nd–2.7	nd	[35]
	Tai Lake, China	14–474	59–271	18–269	19–229	[36]
	Pearl River, China	7.1	67.5	NT	nd	[37]
	Victoria Harbor, China	660–6840	14–2290	NT	NT	[38]
	River, Beijing, China	663.9–2722	NT	NT	NT	[39]
	Huangpu River, China	nd–28.5	nd–0.2	nd–34.2	nd–14.6	[40]
	Yangjie River, China	0.14–4.49	0.11–2.37	0.77–4.32	0.53 5.56	[41]
	Danjiangkou Reservoir, China	2.0–2.9	0.55–0.61	0.87–1.1	0.81 1.2	[42]
	Liuxi River, China	NT	7.06	NT	NT	[43]
	Zhujiang River, China	NT	4.85	NT	NT	[43]
	Shijing River, China	NT	70.4	NT	NT	[43]
	Pearl River Delta, China	NT	4.14–6.62	NT	NT	[44]
	Qingshitan Reservoir, China	50.0–660.13	3.70–3.49	3.49–6.22	4.59–6.06	[45]
	Musi River, India	1553.0–542,452.0	16,148.0–251,137.0	7447.0–5015.7	2262.0–181,609.0	[46]
	Kshipra River, India	640–1460	nd–980	nd	NT	[47]
	Kan River, Iran	NT	NT	9.87	NT	[48]
	Firozabad Ditch, Iran	NT	NT	212.83	NT	[48]
	Po River, Italy	0.65–18.06	NT	1.32–16	NT	[49]
	Arno River, Italy	<1.4–10.88	NT	<1.8–37.5	NT	[49]
	Lambro River, Italy	306	NT	26,200	NT	[50]
	River and Estuaries, Osaka, Japan	nd–510	nd–33	2.6–37	nd–4.4	[51]
	Mitheu River, Kenya	NT	600–2200	500–1300	NT	[52]
	Sagana River, Kenya	NT	nd	200	NT	[52]
	Chania River, Kenya	NT	100–2600	nd	NT	[52]
	Mwania River, Kenya	NT	110	500	NT	[52]
	Kanyuru River, Kenya	NT	nd	200	NT	[52]
	Douro River, Portugal	nd	NT	nd	NT	[53]
	Leça River, Portugal	120	NT	339	NT	[53]
	Titicaca Lake, Perú	NT	NT	85.5–652.7	56.2–63.0	[54]
	East Aurora, USA	NT	NT	nd–360	NT	[55]
Sediment	Suquía River, Argentina	nd–39	nd	NT	NT	[26]
	Jiaozhou Bay, China	nd–33.83	NT	nd–11.52	nd–1.92	[56]
	Yellow River, China	3.07	8.34	32.8	nd	[31]
	Haihe River, China	10.3	32	16	nd	[31]
	Liaohe River, China	3.56	3.32	Nd	nd	[31]
	Qinghe River, China	5.9	49.1	28.3	2.2	[57]
	Huangpu River, China	nd–12.4	NT	nd–3.2	NT	[40]
	Danjiangkou Reservoir, China	1.2–1.9	1.7–2	0.98–1.1	1.2–1.3	[42]
	Pearl River Estuary, China	nd–2.08	50.24–153.06	NT	nd–25.62	[37]
	Pearl River Delta, China	NT	1.88–11.20	0.76–2.42	NT	[44]
	Qingshitan Reservoir, China	1.03–118.11	20.17–722.18	17.48–557.18	4.70–331.82	[45]
	Charmoise River, France	4.9–603	1.6–225	2.9–569	6.6–11	[58]
	Patancheru River, India	nd–3545 *	nd	449–914,044 *	374–102,865 *	[59]
	Musi River, India	890.0–444,916.0	nd–232,918.0	nd–3316.5	4471.0–721,491.0	[46]
	Nakkavagu River, India	0.63	0.14	10	0.064	[60]
	Isakavagu River, India	0.91	0.68	12	nd	[60]
	Mitheu River, Kenya	NT	nd	29.3	NT	[52]
	Sagana River, Kenya	NT	nd	nd	NT	[52]
	Chania River, Kenya	NT	26.6	nd	NT	[52]
	Kanyuru River, Kenya	NT	nd	47.4	NT	[52]
	Titicaca Lake, Perú	NT	NT	950–3010	150–3740	[54]
	Grifn Lake, Switzerland	NT	2.4	2.52	NT	[61]
	Northwest River, USA	<21	NT	<10	NT	[62]
Fish	Guangdong, China	NT	46.64–106.85	27.07–165.15	1.0–34.20	[63]
	Guiyang, China	nd–385.73	nd	nd–16.37	0.30–312.00	[64]
	Hongze Lake, China	NT	NT	15.0–24.0	NT	[65]
	Jining, China	NT	NT	2.08–33.8	nd–3.25	[66]
	Suzhou, China	nd–4.35	NT	nd–33.7	3.7–90.6	[33]
	Pearl River Delta, China	NT	1.95–43.51	1.03–2.16	0.65–1.71	[44]
	Qingshitan Reservoir, China	58.59–968.66	9.15–33.27	12.15–80.26	6.73–102.87	[45]
	Titicaca Lake, Perú	NT	NT	3.4–3.9	3.8–4.8	[54]

OFX: Ofloxacin; NFX: Norfloxacin; ENR: Enrofloxacin; CIP: Ciprofloxacin; nd: The antibiotic was not detected; NT: The antibiotic was not tested; *: Expressed as μg/g organic matter.

**Table 2 antibiotics-11-01487-t002:** Percentages of the similarity of nucleotide and amino acid sequences of the *qnr* genes.

Gene	Percentage of Nuleotide/Amino Acid Similarity (%)						
	*qnrA*	*qnrB*	*qnrC*	*qnrD*	*qnrE*	*qnrS*	*qnrVC*
*qnrA*	100/100	46.33/41.59	59.36/64.22	47.91/46.26	48.99/42.06	58.75/59.63	60.43/61.93
*qnrB*		100/100	48.78/42.99	62.79/64.49	75.81/85.98	48.32/39.72	50/42.99
*qnrC*			100/100	49.77/44.39	46.98/42.52	60.73/60.09	68.65/73.85
*qnrD*				100/100	63.10/65.89	46.51/39.25	50.70/43.93
*qnrE*					100/100	48.37/35.98	49.61/42.06
*qnrS*						100/100	63.93/64.68
*qnrVC*							100/100

*qnrA*, NG_050462.1; *qnrB*, NG_050469.1; *qnrC*, NG_048054.1; *qnrD*, NG_050541.1; *qnrE*, NG_054677.1; *qnrS*, NG_050543.1; *qnrVC*, NG_050551.1. QnrA, WP_012579084.1; QnrB, WP_014386481.1; QnrC, WP_032492368.1; QnrD, WP_012634451.1; QnrE, WP_061586512.1; QnrS, WP_001516695.1; QnrVC, WP_000415714.1.

**Table 4 antibiotics-11-01487-t004:** Detection of *qnr* genes in bacteria isolated from aquatic environments.

Environment	Country	Source	Species Carrying a *qnr* Gene	Location	Reference
Freshwater	Bangladesh	Water	*qnrB*, *qnrS*: *Eschericia coli*	ND	[172]
	Belgium	Water	*qnrS*: ND	ND	[173]
	Brazil	Water	*qnrB, qnrS: Klebsiella pneumonia*	ND	[174]
	Brazil	River Water/Sediment	*qnrB, qnrS*: ND	ND	[175]
	Canada	Wastewater/River Water	*qnrS*: ND	ND	[176]
	Canada, China, Sri Lanka, South Korea, USA	Fish	*qnrA: Aeromonas hydrophila*	ND	[177]
	Chile	Reared fish	*qnrB*: *Citrobacter gillenii*	ND	[151]
	China	Fish	*qnrB, qnrD, qnrS*: *Escherichia coli*	Pl (*qnrS*), ND	[178]
	China	Fish	*qnrA*, *qnrS*: *Aeromonas* spp.	ND	[179]
	China	Fish/River Water	*qnrA*, *qnrB, qnrS*: ND	ND	[180]
	China	Water	*qnrD*, *qnrS*: ND	ND	[181]
	China	Water/Sediment	*qnrS*: ND	ND	[182]
	China	River Water/Sediment	*qnrB, qnrS*: ND	ND	[183]
		Water	*qnrS*: ND	ND	[184]
	China	River Water/Sediment	*qnrA, qnrS: ND*	ND	[185]
	China	Water	*qnrB*: *Klebsiella pneumoniae, Raoultella omithinolytica.*; *qnrS*: *Aeromonas caviae, Aeromonas hydrophila, Aeromonas allosaccharophila, Aeromonas veronii, Escherichia coli, Klebsiella pneumoniae, Enterobacter hormaechei, Leclercia adecarboxylata, Enterococcus faecalis*	Pl (*qnrS*), ND	[186]
	China	Fish/Shrimp	*qnrA*, *qnrD*: ND	ND	[13]
	China	Shrimp	*qnrD*: ND	Pl	[187]
	China	Shrimp	*qnrA*: ND	ND	[188]
	China	River Water	*qnrS*: ND	ND	[189]
	China	Lake Water	*qnrA, qnrB, qnrD, qnrS*: ND	ND	[190]
	China	Lake Water	*qnrA, qnrB, qnrD, qnrS*: ND	ND	[191]
	China	Wastewater/River Water	*qnrA*: *Enterobacter* sp., *Proteus* sp., *Citrobacter* sp.; *qnrB: Klebsiella* sp., *Enterobacter* sp., *Proteus* sp., *Shigella* sp., *Citrobacter* sp. *qnrS: Klebsiella* sp., *Escherichia coli, Enterobacter* sp.;	Pl, Cr	[192]
	China	Wastewater/River Water	*qnrC, qnrD*: ND	Pl	[39]
	China	Wastewater/River Water	*qnrA, qnrB, qnrS*: ND	ND	[193]
	China	Eel, Pond Water	*qnrA, qnrB, qnrS*: ND	ND	[194]
	China	Water	*qnrB: Salmonella typhi, Salmonella enteriditis, Salmonella typhimurium*	ND	[195]
	China	River Water/Sediment	*qnrA, qnrB, qnrD, qnrS*: ND	ND	[196]
	China	River Water	*qnrD*: ND	ND	[197]
	China	Water	*qnrS*: ND	ND	[198]
	China	Water	*qnrA, qnrD, qnrS*: ND	ND	[199]
	China	Water	*qnrB, qnrS*: *Escherichia coli*	Pl	[200]
	China	Water	*qnrD, qnrS*: ND	ND	[201]
	China	River Water	*qnrD, qnrS*: ND	ND	[202]
	China	River Water/Sediment	*qnrA*: ND	ND	[203]
	China	Water/Sediment	*qnrA, qnrB, qnrD, qnrS*: ND	ND	[204]
	Egypt	Fish	*qnrA, qnrB, qnrS*: *Edwarsiella tarda*	ND	[205]
	France	Water	*qnrS*: *Aeromonas punctata, Aeromonas media*	Pl	[206]
	India	Fish	*qnrS*: *Aeromonas hydrophila*	Pl	[207]
	India	Sediment	*qnrB, qnrD, qnrS, qnrVC*: ND	ND	[69]
	India	River Sediment	*qnrS:* ND	ND	[208]
	India	River Water	*qnrD, qnrS, qnrVC*: ND	ND	[59]
	India	Water	*qnrS*: *Acinetobacter* sp., *Pseudomonas* sp., *Aeromonas* sp., *Brevibacterium* *frigoritolerans*	Pl	[209]
	India	Water/Sediment	*qnrA, qnrB*: ND	ND	[210]
	India, Sri Lanka	Water	*qnrS*: ND	ND	[211]
	Iran	Water	*qnrA, qnrB, qnrS*: *Escherichia coli*	ND	[212]
	Iraq	Water	*qnrA: Escherichia coli*	ND	[213]
	Ireland	River Water	*qnrS:* ND	ND	[214]
	Italy	River Water	*qnrS:* ND	ND	[215]
	Italy	Water	*qnrS:* ND	ND	[216]
	Italy	River Water	*qnrS*: ND	ND	[217]
	Italy	River Water	*qnrS*: ND	ND	[218]
	Japan	River Water	*qnrS*: *Escherichia coli*	Pl	[51]
	Korea	Fish	*qnrS*: *Aeromonas* sp.	Pl	[219]
	Mexico	Sediment	*qnrB:* ND	Pl	[220]
	Mexico, USA	Sediment	*qnrA, qnrB, qnrS:* ND	ND	[221]
	Poland	Water	*qnrD*: *Eschericia coli, Acinetobacter* sp., *Acinetobacter johnsonii, Acinetobacter guillouiae, Aeromonas* sp., *Bacillus* sp., *Pseudomonas* sp., *Cronobacter* sp., *Acidovorax* sp., *Hydrogenophaga* sp., *Kurthia* sp., *Providencia sp., Psychrobacter* sp., *Shigella* sp., *Vibrio* sp. *qnrS: Sphingobacterium* sp., *Pedobacter* sp., *Eschericia coli, Acinetobacter* sp., *Acinetobacter johnsonii, Aeromonas* sp., *Bacillus* sp, *Kurthia* sp., *Shigella* sp.	Pl	[222]
	Portugal	Water	*qnrA*: *Escherichia coli*	ND	[223]
	Portugal	Water	*qnrS*: ND	ND	[224]
	Portugal	Water	*qnrA, qnrB, qnrS*: ND	ND	[225]
	Portugal	Water	*qnrVC*: *Aeromonas hydrophila, Pseudomonas* sp., *Escherichia coli, Aeromonas* sp.	ND	[226]
	Spain	Biofilm	*qnrB*: *Klebsiella oxytoca; qnrS*: *Aeromonas* sp.	Pl	[227]
	Spain	Biofilm	*qnrS*: ND	ND	[228]
	Spain	Sediment	*qnrB*: *Citrobacter freundii; qnrS*: *Aeromonas* sp., *Raoutella terrígena*	Pl	[227]
	Spain	Sediment	*qnrS*: ND	ND	[229]
	Spain	River Water	*qnrS*: ND	ND	[230]
	Spain	Sediment	*qnrB: Escherichia coli, Raoultella ornithinolytica, Enterobacter cloacae*	Pl	[231]
	Spain	Water	*qnrS*: ND	ND	[22]
	Spain	Water	*qnrS*: ND	ND	[232]
	Spain	Water	*qnrA, qnrS*: ND	ND	[233]
	Spain	Water	*qnrS*: ND	ND	[234]
	Spain	Water/Sediment	*qnrA, qnrS*: ND	ND	[235]
	South Africa	Fish	*qnrB*: *Aeromonas veronii, Aeromonas caviae, Aeromonas hydrophila, Aeromonas jandaei qnrS*: *Aeromonas veronii, Aeromonas hydrophila, Aeromonas jandaei*	ND	[236]
	Switzerland	Lake Water	*qnrS: Aeromonas allosaccharophila*	Pl	[164]
	Switzerland	Water	*qnrS*: *Escherichia coli*	Pl	[237]
	Thailand	Shrimp/Pond Water	*qnrVC: Vibrio parahaemolyticus*	ND	[238]
	Thailand and Vietnam	Water	*qnrB*: *Brevundimonas diminuta, Blastobacter aggregatus, Janibacter anophelis*; *qnrS*: *Escherichia coli*	ND	[239]
	United Kingdom	River Water	*qnrS*: ND	ND	[240]
	United Kingdom	River Water	*qnrS*: ND	ND	[241]
	United States	River Water	*qnrA*: ND	ND	[242]
Seawater	Antarctica	Sediment	*qnrS*: ND	ND	[243]
	Australia	Water	*qnrS*: ND	ND	[244]
	Brazil	Water/Sand	*qnrA, qnrB, qnrS*: ND	ND	[245]
	Chile	Water/Sediment	*qnrA: Alcanivorax* sp., *Arcobacter* sp., *Arthrobacter* sp., *Kytococcus* sp., *Marinobacter* sp., *Microbacterium* sp., *Pseudomonas* sp., *Rhodococcus* sp.; *qnrB: Kytococcus* sp., *Marinobacter* sp., *Rhodococcus* sp., *Actinobacterium* sp., *Cellulophaga* sp., *Flavobacteriaceae, Erythrobacter* sp., *Tsukamurella* sp.; *Rhodococcus* sp.; *Marinobacter* sp.; *Kylococcus* sp.; *qnrS: Arcobacter* sp., *Arthrobacter* sp., *Marinobacter* sp., *Pseudomonas* sp., *Rhodococcus* sp., *Cellulophaga* sp., *Erythrobacter* sp., *Dietzia* sp., *Microbacter* sp.	Pl, Cr	[246,247]
	China	Fish	*qnrA, qnrC*: ND	ND	[178]
	China	Sediment	*qnrA: Escherichia coli, Proteus mirabilis, Providencia stuartii, Klebsiella pneumoniae*	Pl, ND	[248]
	China	Sediment	*qnrS:* ND	ND	[249]
	China	Fish	*qnrA, qnrB, qnrD, qnrS*: ND	ND	[250]
	China	Sediment	*qnrA,qnrB,qnrD,qnrS:* ND	ND	[251]
	China	Water/Sediment	*qnrB,qnrS*: ND	ND	[252]
	China	Water	*qnrA*: *Shewanella algae; qnrB*: *Citrobacter freundii; qnrD*: *Proteus vulgaris; qnrS*: *Enterobacter* sp., *Klebsiella pneumoniae, Pseudoalteromonas* sp., *Pseudomonas* sp.	Pl, Cr	[253]
	China	Water/Sediment	*qnrS*: ND	ND	[254]
	Egypt	Water	*qnrA*: *Klebsiella pneumoniae, Enterobacter cloacae, Citrobacter koseri, Proteus mirabilis, Shewanella putrefaciens; qnrB*: *Klebsiella pneumoniae, Citrobacter koseri*, *qnrS: Klebsiella pneumoniae, Aeromonas hydrophila, Enterobacter cloacae, Escherichia coli, Citrobacter* sp., *Pasteurella* sp.	ND	[255]
	Italy	Water	*qnrS:* ND	ND	[216]
	Italy	Water/Sediment	*qnrA: Shewanella algae; qnrVC: Vibrio anguillarum*	Pl	[256]
	Portugal	Fish	*qnrB*: *Leclercia adecarboxylata*	ND	[257]
	Portugal	Clams/Oysters	*qnrA, qnrB, qnrS*: ND	Pl	[258]
	Singapore	Water/Sediment	*qnrA*: ND	ND	[259]
	South Korea	Fish Farm Effluent	*qnrS:* ND	Pl	[260]
	South Korea	Fish Farm Effluent	*qnrD: Psychrosphaera; qnrS:* ND	ND	[261]
	Turkey	Fish	*qnrS*: *Pantoea agglomerans*	ND	[262]
	United States	Fish	*qnrS*: *Vibrio* sp.	ND	[263]

Cr, Chromosomal; Pl, Plasmid; ND, Not Determined.

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
