# Peer review of "Aquatic Environments as Hotspots of Transferable Low-Level Quinolone Resistance and Their Potential Contribution to High-Level Quinolone Resistance"

_antibiotics, 2022, doi:10.3390/antibiotics11111487_

Round 1

Reviewer 1 Report

The attention this review brings to the importance of low-level resistance in environmental populations, as well as to current knowledge gaps on that topic, is both timely and constructive. The authors do an excellent job of presenting the scope of the problem of antibiotic resistance, explaining the mechanisms, and summarizing the state of the science. The quality of writing is good, apart from minor stylistic issues (e.g., excessive use of transition words and phrases; long compound/run-on sentences; and some choppy two-sentence paragraphs).

I do, however, have a major concern. The authors' argument that qnr genes serve as important precursors to high-level resistance seemed like hand-waving to me. The first pass at it in the Introduction ended with "as previously discussed" (Line 62), presumably in reference to the Abstract (which, in turn, included the following unintelligible statement: "these genes can act favoring the appearance of chromosomal mutations in the sequences that code for the target proteins of quinolones leading to high-level quinolone resistance"). Elsewhere, the general take-away seemed to be that high-level resistance must evolve from low-level resistance, which is theoretically inaccurate/misleading. Especially given that the authors acknowledge that clinically resistant strains do not have qnr genes, I would suggest doing at least one of two things: (1) be more explicit about how qnr genes and/or the enzymes they produce can mutate into ones that provide high-level resistance... or (2) re-frame the argument in terms of how low-level resistance can exacerbate the problem of high-level resistance by helping to maintain bacterial diversity and by supplementing other pathways of resistance. Lines 376-386 provide some good fodder for the latter approach.

Reviewer 2 Report

The main issues of this manuscript are as following:

1、 The second part ”antimicrobial residues in aquatic environments” should mainly focus on the quinolone antibiotics.

2、 Figure 1, please explain what the elements stand for in the figure.

3、 Table 2, please explain what the numbers in the brackets mean.

4、 The 3.3 to 3.6 parts are not mechanisms of quinolone resistance, so they should be dependent from part 3. The structure should be reorganized again.

5、 The low-level and high-level resistance should be defined clearly and compared.

6、 Lin 500, this figure should be figure 2.

7、 The last part of “conclusions and perspectives” should be simplified and not contain references. Moreover, the perspectives should be clearly pointed out.

8、 Many sentences in this manuscript are too long and have grammatical mistakes, such as line64-66, line 208-212, line 637-642.

9、 There are too many small paragraphs in the manuscript, for example, part 3.5, authors should reorganize the structure and summarize better.

Round 2

Reviewer 1 Report

I commend the authors on their hard work. They have addressed my major concern and corrected most of the wording and style issues of the previous draft. Some minor issues remain, however, that I believe are worth addressing and can be addressed quickly and easily. These are as follows:

*Paragraphs comprising three sentences or less should be combined into paragraphs of at least four sentences. This will save space and reduce the "choppiness" of the manuscript.

*In some places, the insertions/revisions have created unnecessary blank spaces. These should be deleted.

*In Lines 513-514, the phrase "can persist for extended periods of time" should be replaced with "are extensive."

*In Line 538, either the "a" should be deleted or "periods" should be made singular.

*In Line 622, "in" should be "is."

Author Response

PLease, see the attachment

Reviewer 2 Report

There are still some issues should be addressed in the revised manuscript as following:

1. In the second part of ’Antimicrobial residues in aquatic environments‘, the title should be 'quinolone residues in aquatic environment' and more data  should be added, ie. add a table to summerize the concentrations of quinolones in aquatic environment reported in the pubulished papers.

2. In the part of "Mechanisms of quinolone resistance', only 'Acquired low-level resistance to quinolones' was described, how about high level-resistance to quinolones was acquired? 
